# Preparation of Activated Carbon from Coffee Waste as an Adsorbent for the Removal of Chromium (III) from Water. Optimization for an Experimental Box-Behnken Design

**Adolfo Eduardo Obaya Valdivia \*** , **Carlos Montaño Osorio and Yolanda Marina Vargas Rodríguez**

Departamento de Ciencias Químicas, Facultad de Estudios Superiores Cuautitlán-Universidad Nacional Autónoma de México, Campo No. 1. Av. 1 de mayo, Sta. María Las Torres, Cuautitlán Izcalli, Estado de México, México C.P. 54740, Mexico; camo.osorio10@gmail.com (C.M.O.); ym_vargas@yahoo.com.mx (Y.M.V.R.)
\* Correspondence: obaya@unam.mx; Tel.: +52-55-56239441

**Abstract:** Coffee grounds are an organic waste consisting of the ground, roasted and processed grain for the extraction of coffee, being of great volume the waste obtained, which, if not treated and preserved for a long time, emanates unpleasant aromas and becomes an optimal means for the proliferation of flies, and other pests. Activated carbon has the characteristic of being a material that has a large surface area; because of this, it is used in adsorption, which refers to the retention of atoms, ions, and molecules on its surface. In this paper, the production of activated carbon is presented by means of the physicochemical activation of coffee waste by calcining at 900 °C and subsequent activation with sulfuric acid, as well as the Box–Behnken design of three factors for chromium (III) adsorption optimization. It is determined that the optimal conditions for performing the adsorption are pH = 3 and a contact time of 140 min. According to the analysis of variance for the experimental design, it is determined that the initial chromium formation is not significant for the adsorption process. Under the optimal adsorb conditions of 96%, an application is given to the erasure of coffee for the production of activated carbon, which serves as the adsorbent agent applied to the removal of chromium (III) aqueous. The area of activated carbon obtained is 13657.89 ± 251.09 $m^2/g$ determined by the acetic acid adsorption isotherm method.

**Keywords:** adsorption; activated carbon; chromium (III), Box–Behnken design

## 1. Introduction

From various industrial activities such as electroplating, painting, batteries, textile tanning, among others, wastewater is generated with high levels of heavy metal concentration which accumulate in aquifers [1] (Buszman & Zdybel, 2006), whose presence constitutes a pollution problem such as the residual chromium of the aforementioned industries. This can be poured as hexavalent chromium (dichromates and chromates) since it has a high oxidation standard potential, reacts with the organic matter by oxidizing it and consequently, it is reduced to trivalent chromium ($Cr^{3+}$), which is why the predominant species in aqueous residues is the $Cr^{3+}$ ion. These effluents should be treated to minimize their concentration, considering the great environmental impact they generate [2].

Within the treatments for the removal of chromium can be found REDOX reactions, adsorption and precipitation, as well as biological reactions (enzymatic REDOX reactions carried out by microorganisms and phytoremediation, as well as exchange methods [3], but not all are economical, the most accessible

being adsorption, where an adsorbent agent is used, contacting the aqueous medium from which the chromium will be removed, for this it is necessary that the material has a large area interfacial.

Adsorption phenomena are of great importance in the development of operations and processes in the chemical industry, as well as in the academic and research fields. That is why heavy metal removal techniques have been developed using different solids such as clinopthyloid [4] diatomite [5], lignin [6] and metal oxides [7]. The advantage of using activated carbon as an adsorbent is the low generation of waste, the easy recovery of metals and the possibility of reusing it [8], it is even used in gas purification because it has a high Surface area and economically is low cost [9]. The term "activated carbon" refers to carbons produced from carbon-rich materials, through various forms of activation, chemical, physical, and physicochemical [10]. For the preparation of activated carbons, the material is calcined (carbonization) and then undergoes the activation process, which leads to increase porosity and adsorbent capacity of the carbonized material by oxidation treatments of the functional groups of the surface of the solid. Various oxidation procedures are used to introduce oxygenated groups on the surface of the coal, by gas ($O_2$, $CO_2$, $N_2O$), or by using aqueous solutions ($H_2O_2$, $HClO_4$ or $HNO_3$) [11] is by that activated carbon is a widely used adsorbent material, since it can be prepared from organic raw material, such as organic waste of various types.

The coffee grounds (coffee residues) are organic waste consisting of the grain of ground and roasted coffee, which after the extraction of the coffee is discarded, so large masses of it are generated. Such residue, if stored, becomes an average potential for growing mainly of fungi, in addition to emanating unpleasant odors by decomposition of the different organic matter it possesses. Due to its high content of organic matter, it is possible to make activated carbon from such wastes. Given the above, the objective arises of this research work, which is the use of coffee eraser for the preparation of activated carbon through a physicochemical process (calcined and acid treatment) for the removal of chromium from water as well as the study of the adsorption system by designing three-level, three-factor Box–Behnken experiments for process optimization by building the system's response surface and determining the mathematical model that describes the percentage of adsorption, as well as the determination of the surface area of the carbon obtained by the acetic acid adsorption technique.

## 2. Materials and Methods

Equipment and Materials Analytical grade reagents were used: Sulfuric acid (J. T. Baker, Phillipsburg, NJ, USA, 99.8%), Chromium chloride (lll) (J. T. Baker, 99%), disodic salt of EDTA (Merck, Kenilworth, NJ, USA, 99.98%). The absorbance readings were performed on a Perkin Elmer Soft Lambda 18 spectrophotometer (Pekin Elmer, Waltham, MA, USA). Activated carbon was made in a muffle furnace (Vulcan Muffle Furnace 9592, Cole-Parmer, Vernon Hills, IL, USA).

### 2.1. Preparation of Activated Carbon

300 g of dry coffee residues are calcined in the muffle furnace at 900 °C for 3 h; then the obtained carbon is contacted with 20 mL of 10% sulfuric acid and heated to 65 °C for one hour. The carbon is then filtered and washed with deionized water and dried on a stove at 110 °C for an hour. The dry solid is already activated carbon. Finally, it is finely ground in a mortar, made this last step, the carbon can already be used as an adsorbent agent.

### 2.2. Experimental Design

The experiments were planned by defining the factors and levels as specified in Table 1:

**Table 1.** Experimental design for surface response construction.

| Factor/Level | −1 | 0 | +1 |
|---|---|---|---|
| [Cr (lll)], mol/L | 0.01 | 0.055 | 0.1 |
| pH | 1 | 2 | 3 |
| Time (min) | 100 | 120 | 140 |

The experiments were carried out by placing 20 mL aliquots for each system by adding 1 g of activated carbon obtained from the coffee erase. The pH of the solutions was adjusted by adding small droplets of the hydrochloric acid analytical reagent.

The choice of the pH range was determined based on the diagram of predominance zones for Cr (III) which begins to precipitate at pH > 3.2 for the chromium concentrations used, which can be seen in Figure 1.

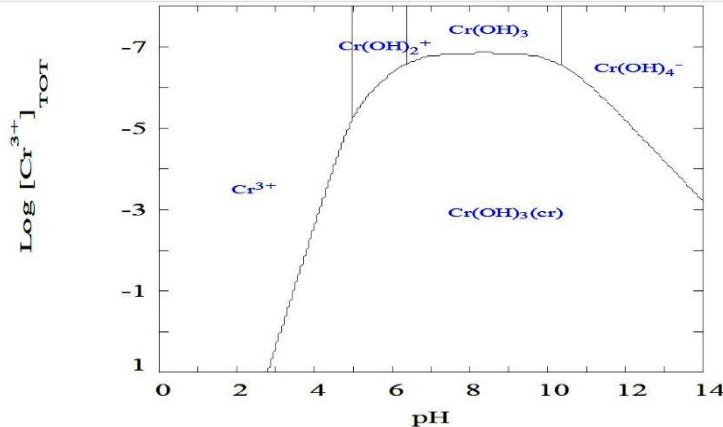

**Figure 1.** Chromium (III) Predominance Existence Diagram.

The experiments are performed according to the matrix in Table 2.

**Table 2.** Matrix experimental design Box–Behnken.

| Experiment | Factor 1: [Cr(lll)] (mol/L) | Factor 2: pH | Factor 3: Time (min) |
|---|---|---|---|
| 1 | −1 | −1 | 0 |
| 2 | 1 | −1 | 0 |
| 3 | −1 | 1 | 0 |
| 4 | 1 | 1 | 0 |
| 5 | −1 | 0 | −1 |
| 6 | 1 | 0 | −1 |
| 7 | −1 | 0 | 1 |
| 8 | 1 | 0 | 1 |
| 9 | 0 | −1 | −1 |
| 10 | 0 | 1 | −1 |
| 11 | 0 | −1 | 1 |
| 12 | 0 | 1 | 1 |
| 13 | 0 | 0 | 0 |
| 14 | 0 | 0 | 0 |
| 15 | 0 | 0 | 0 |

In each experiment, it was determined the initial and final absorbance to perform the percentage of chromium removal according to Equation (1):

$$\% \, Removal = 1 - \frac{A}{A_o},$$ (1)

Determination of the surface area of activated carbon using the acetic acid adsorption isotherms method. Develop 10 systems of 25 mL with different acetic acid concentrations described in Table 3 from dilutions of a standard acetic acid solution 1.4097 M.

**Table 3.** Acetic acid concentrations for isothermal adsorption.

| System | a | b | c | d | e | f | g | h | i | j |
|---|---|---|---|---|---|---|---|---|---|---|
| Co Hac (M) | 0.0563 | 0.1127 | 0.1691 | 0.2255 | 0.2819 | 0.3383 | 0.3947 | 0.4511 | 0.5074 | 0.5638 |

The letters a, b, c, d, e, f, g, h, i, j are the concentrations of acetic acid for the determination of isotherms.

Ten mL of each solution was taken and poured into a bottle, then 1 g was added of the activated carbon obtained. The bottle was covered tightly and left in agitation for 3 h under a constant temperature bath at 25 °C. Afterward, the solutions were filtered and aliquots of 5 mL were titrated with a standardized solution of sodium hydroxide 0.0228 M. The process was performed in triplicate. The amount of acetic acid moles adsorbed by activated carbon mass is determined by Equation (2):

$$\frac{n_{HAc\ ads}}{m} = \frac{V\left(C_o - C_{eq}\right)}{m},$$ (2)

where: $n_{HAc\ ads}$ = activated carbon absorbed [mol]; $m$ = activated carbon weight [g]; $V$ = Volume of the aliquot valued [L]; $C_o$ = Initial acetic acid concentration Acetic acid $\left[\frac{mol}{L}\right]$; $C_{eq}$ = balance concentration determined by NaOH titration $\left[\frac{mol}{L}\right]$.

## 3. Results and Discussion

Once the experiments were completed, we proceeded to fill the experimental matrix with the response (% chromium removal) to obtain the response surfaces from Figure 2.

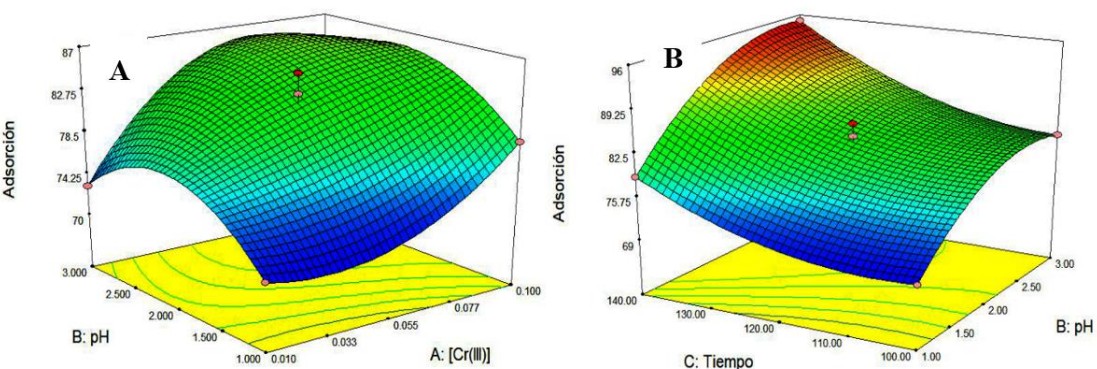

**Figure 2.** Response surfaces for chromium adsorption (III). (**A**) Surface in the plane pH-[Cr (lll)], (**B**) in the plane time-pH.

As can be seen in Figure 2, a maximum of the response surface (A) is found, the maximum percentage of chromium removed is 97%, at an initial concentration of Cr (III) of 0.1 M and a pH of 3. Likewise, it is observed that the pH has an optimum of 2, since if the medium is strongly acidic (pH = 1), the protons compete for the active sites with the chromium ions (III), as shown in Figure 3, the species Chromium that adsorbs is $Cr^{3+}$ in the pH range studied.

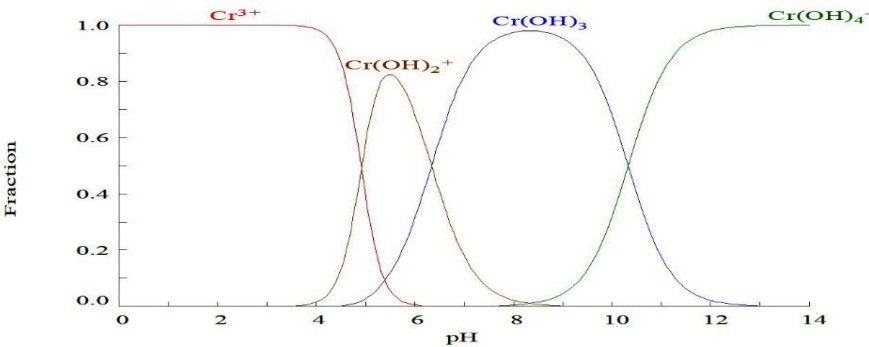

**Figure 3.** Cr (III) Species Distribution Diagram.

In Figure 1, it is observed that the time does not present a maximum, since the more the contact time, there is a higher percentage of removal of chromium ion in the solution.

The mathematical model in coded terms of the response surface is as follows:

$$\%Removal = 83 + 4A + 6.75B + 5.75C + 0.5AB - 2.50AC + 1.25BC - 1.62A^2 \\ -4.87B^2 + 3.12C^2 - 5.25A^2C + 0.5AB^2, \tag{3}$$

Therefore, the factors with the greatest significance in chromium removal are pH (B), followed by contact time (C), and finally, the initial chromium concentration (A); on the other hand, the interactions between the factors are weak since their coefficients are Low. For the assertion that the mathematical model correctly describes the experimental results, the following residual chart is analyzed:

As shown in Figure 4, the residuals are above the normal probability line; therefore, the mathematical model is correct and adequately describes the results.

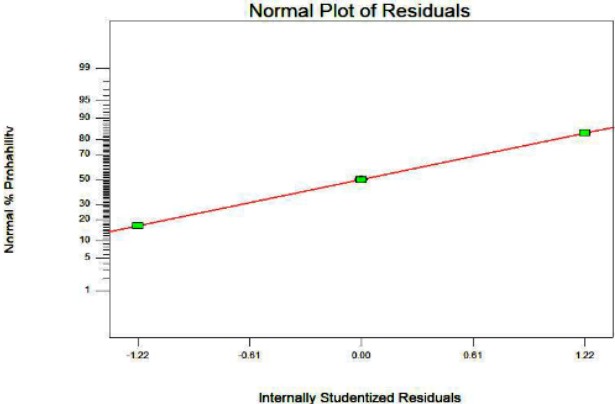

**Figure 4.** Graph of residuals for the mathematical model of experimental design.

Finally, the optimal conditions for the removal of chromium on the activated carbon.

According to the experimental design, the optimal conditions for chromium removal are presented in Table 4.

**Table 4.** Optimal conditions obtained by the Box–Behnken design for chromium absorption (III).

| Factor | Level |
| --- | --- |
| [Cr(lll)] (mol/L) | 0.1 |
| pH | 2.14 |
| Time (min) | 140 |

For the determination of the surface area, the amount of adsorbed acetic acid for each gram of activated carbon was determined, as shown in Table 5.

**Table 5.** Acetic acid adsorption results on activated carbon.

| System | a | b | c | d | e | f | g | h | i | j |
|---|---|---|---|---|---|---|---|---|---|---|
| Co Hac (M) | 0.0564 | 0.1128 | 0.1692 | 0.2256 | 0.2819 | 0.3383 | 0.3947 | 0.4511 | 0.5075 | 0.5639 |
| Ceq HAc (M) | 0.0019 | 0.0040 | 0.0063 | 0.0089 | 0.0119 | 0.0152 | 0.0190 | 0.0234 | 0.0285 | 0.0344 |
| $n_{HAc\ ads}/m$ (mol/g) | 0.0005 | 0.0011 | 0.0016 | 0.0022 | 0.0027 | 0.0032 | 0.0038 | 0.0043 | 0.0048 | 0.0053 |

The letters represent the levels of acetic acid concentration and the amount of moles assigned per gram of activated carbon.

Values fit the following adsorption isotherm models in their linearized equation.
Langmuir's adsorption isotherm:

$$\frac{C_{eq}}{\left(\frac{n_{HAc\ ads}}{m}\right)} = \frac{1}{\left(\frac{n_{HAc\ ads}}{m}\right)_{max} b} + \frac{1}{\left(\frac{n_{HAc\ ads}}{m}\right)_{max}} C_{eq}, \tag{4}$$

Freundlich's adsorption isotherm:

$$ln\left(\frac{n_{HAc\ ads}}{m}\right) = lnK + \frac{1}{n} lnC_{eq}, \tag{5}$$

Braunauer-Emmett-Teller (BET) adsorption isotherm:

$$\frac{C_{eq}}{\left(\frac{n_{HAc\ ads}}{m}\right)*\left(C_o - C_{eq}\right)} = \frac{1}{\left(\frac{n_{HAc\ ads}}{m}\right)_{max} C} + \frac{C-1}{\left(\frac{n_{HAc\ ads}}{m}\right)_{max} C} * \left[\frac{C_{eq}}{C_o}\right], \tag{6}$$

Figures 5–7 show the linear regression plots for the determination of the adsorption model data.

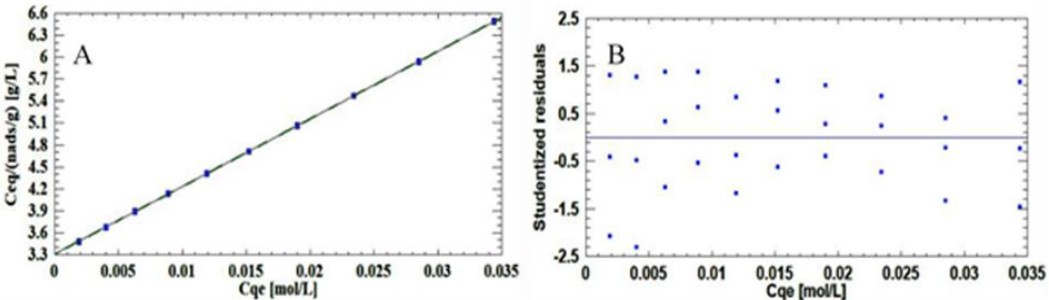

**Figure 5.** Langmuir model plot (**A**) residuals plot (**B**) to the Langmuir model.

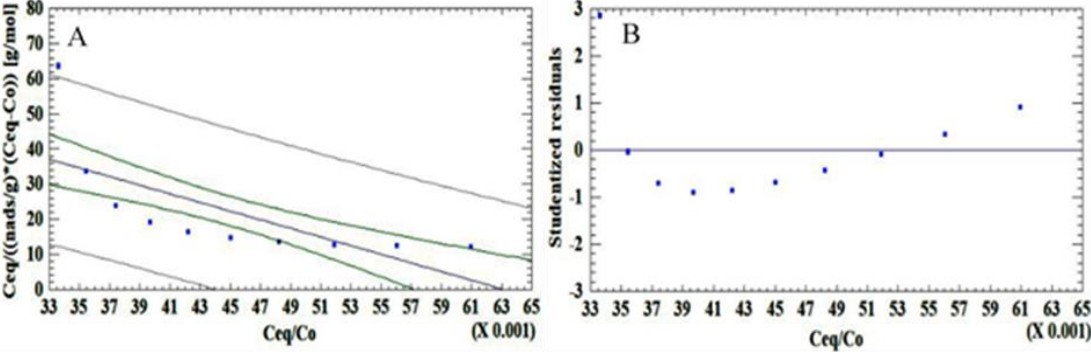

**Figure 6.** BET model (**A**) residuals plot (**B**) Freundlich isotherm.

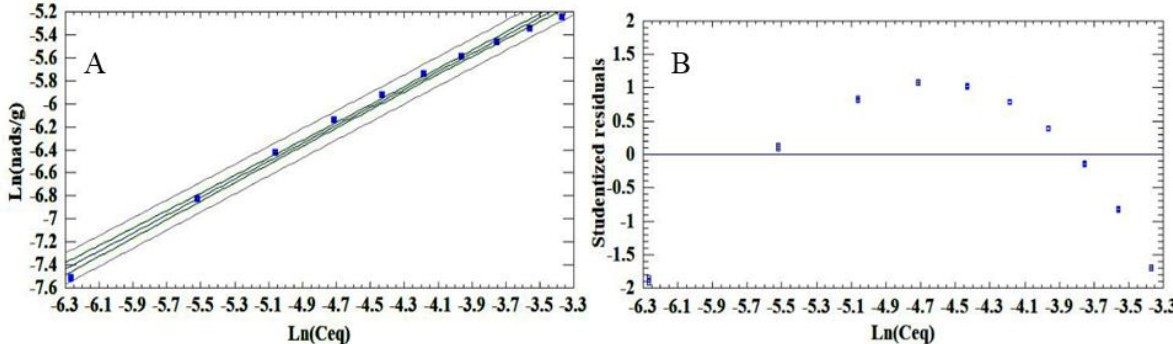

**Figure 7.** Langmuir model plot (**A**) residuals plot (**B**).

Table 6 shows the linear regression for the different mathematical adsorption models.

**Table 6.** Linear regression statistics for mathematical adsorption models.

| Isotherm. | Linear Regression Models | $R^2$ | Residuals |
|---|---|---|---|
| Langmuir | $\frac{C_{eq}}{\left(\frac{n_{ads}}{g}\right)} = 3.3032(\pm0.0013) + 92.6907(\pm0.0729)C_{eq}$ | 0.9988 | Homocedastic |
| Freundlich | $Ln\left(\frac{n_{ads}}{g}\right) = -2.5053(\pm0.0584) + 0.7831(\pm0.0127)lnC_{eq}$ | 0.9962 | Heterocedastic |
| BET | $\frac{C_{eq}}{\left(\frac{n_{HAc\ ads}}{m}\right)\times\left(C_o-C_{eq}\right)} =$ <br> $77.7626(\pm10.9234) - 1232.53(\pm238.11)\left[\frac{C_{eq}}{C_o}\right]$ | 0.4889 | Heterocedastic |

Based on the $R^2$ values and the arrangement of the residuals, it is concluded that the data conform to the Langmuir model, which is corroborated by the following hypothesis test: Ho: There is a linear adjustment. H1: There is no linear adjustment.

Accepting the null hypothesis if the null hypothesis, the *p*-value of the ANOVA table with a lack of adjustment is less than 0.05.

As shown in Table 7, the *p*-value of the ANOVA table is greater than 0.05, so the Langmuir linear model describes the behavior of the data with a 95% confidence level. From the regression parameters of the Langmuir isotherm, the surface area of the activated carbon obtained is calculated:

$$\frac{1}{\left(\frac{n_{HAc\ ads}}{m}\right)_{max}} = 92.6907\frac{g}{mol\ HAc},$$

$$\left(\frac{n_{HAc\ ads}}{m}\right)_{max} = 0.0108\frac{mol\ HAc}{g},$$

The Surface area is calculated with Equation (7):

$$Surface\ area = \left(\frac{n_{HAc\ ads}}{m}\right)_{max}N_A\sigma, \qquad (7)$$

where $N_A$ = Avogrado number; $\sigma$ = Molecule area adsorbed, Acetic acid $21\times10^{-19}\frac{m^2}{molecula}$ [12]. $Surface\ area = \left(0.0108\frac{mol\ HAc}{g}\right)\times\left(\frac{6.022\times10^{23}\ molecule\ HAc}{mol\ HAc}\right)\times\left(21\times10^{-19}\frac{m^2}{molecule\ HAc}\right)$, $Surface\ area = 13657.89\frac{m^2}{g\ activaved\ carbon}$.

The area of activated coals is variant according to the preparation method and the source of the coal, for example, the wood carbons have a surface area of 800 m²/g, while coals seamlessly reach surface areas of 2000 m²/g [13–15]. In this work, activated carbon was obtained, which has a surface area of 13657.89 ± 251.09 m2/g from the use of coffee residues through calcining activation and subsequent acid treatment.

**Table 7.** ANOVA lack of fit test.

| Source | SS | Gl | MS | F-Value | *p*-Value |
|---|---|---|---|---|---|
| Modelo | 3418.71 | 1 | 3418.71 | 26.79 | 0.0000 |
| Residual | 3572.71 | 28 | 127.597 | | |
| Lack of fit test | 3572.67 | 8 | 446.584 | 230314.53 | 0.0452 |
| Error | 0.0387804 | 20 | 0.00193902 | | |
| Total (Corr.) | 6991.42 | 29 | | | |

## 4. Conclusions

It was possible to obtain carbon activated by heat treatment and subsequent acid, from the use of the coffee erases, thus giving a utility to this product, which is discarded, also the experimental design Box–Behnken of three levels with three repetitions to the center, to perform the repetition of the response, from which the optimal adsorption conditions are obtained for the factors studied, these are an initial chromium concentration of 0.1 M, a pH of 2.14 and a contact time of 140 min of the surface area of activated carbon the acetic acid adsorption technique, obtaining a surface area of 13657.89–251.09 m$^2$/g. In this work, the use of the coffee bean is obtained to obtain activated carbon and its study as adsorbent material of an ion as toxic and polluting as chromium is.

**Author Contributions:** C.M.O. was responsible for conducting the experimental design and performed the experiments as well as the use and interpretation of the Software Design Expert. A.E.O.V. made the original draft, and was in charge of the writing. Y.M.V.R. was responsible for the administration of the project and the acquisition of funds, and also determined the area of activated carbon by the method of adsorption isotherms. All authors have read and agreed to the published version of the manuscript.

**Funding:** This research was funded with funds from the UNAM-PAPIIT IN229119 project "Study of the catalytic activity of iron-nanotubule oxide surfaces of halloysite, in the degradation of emerging pollutants".

**Acknowledgments:** This work was supported by UNAM-PAPIIT IN229119.

**Conflicts of Interest:** The authors declare no conflict of interest.

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
