# Peer review of "Preparation of Activated Carbon from Coffee Waste as an Adsorbent for the Removal of Chromium (III) from Water. Optimization for an Experimental Box-Behnken Design"

_chemistry, doi:10.3390/chemistry2010002_

Round 1

Reviewer 1 Report

The authors present the fabrication and characterization of activated carbon from coffee waste towards heavy metal removal.

However, while the experiments are systematic and well conducted, the manuscript is riddled with many English errors (i.e., run-on sentences, sentence fragments, lack of appropriate punctuation marks, incorrect subject-verb agreements, etc.) that make it difficult to read. For example, the first sentence of the abstract compacts 2-3 sentences together, which makes it difficult to understand without several readthroughs of that one sentence for clarity. Due to this, the work needs a thorough review before consideration for publication.

The authors present the fabrication and characterization of activated carbon from coffee waste towards heavy metal removal. They discuss the performance of the manufactures activated carbon with experimental design with solution pH, chromium (III) concentration, and adsorption time as varied parameters. However, while the experiments are systematic and well conducted, the manuscript is riddled with many English errors (i.e., run-on sentences, sentence fragments, lack of appropriate punctuation marks, incorrect subject-verb agreements, etc.) that make it difficult to read. Due to this, the work needs a thorough review with the following suggestions before consideration for publication. 1. Fixing many grammatical errors, including but not limited to: Line 12-15: Run-on sentences make the abstract unreadable. Line 18: better wording would be "..presented by means of the physicochemical activation of coffee waste..." Line 20: chromium and (III) should be together Line 21: "the initial concentration of chromium..." is misplaced and should be moved, possibly as a separate sentence. Line 29: The series includes electroplate and electroplating redundancy. Does whiskies refer to singular whisky or alcohol in general? Should "in others" be "and others"? Line 31: "susceptible" should better refer to the water being contaminated, but it is used to describe heavy metals without the use of appropriate commas. Line 33: "Mainly from various industries" is extremely vague. Line39-44: Run-on sentences and several missing puctuation marks make the paragraph unreadable. 2. The "mufla" in Line 74 should be refered to as a box furnace or kiln. The model of the box furnace should be included. 3. The parameters from Table 1 are fine, but possibly have the pH range a bit wider. If not, provide justification of the small range of pH 1-3. And the caption of Table 2 is mismarked as Table 1. 4. With pH being determined as the parameter with the strongest correlation to chromium removal from the experimental design, would there be a futher investigation with expanding the pH range? 5. I don't quite understand the referal o a the caption for Figure 3, 4, and 5. Possibly rewrite them as full sentences.

Author Response

Point 1:  Fixing many grammatical errors, including but not limited to: Line 12-15: Run-on sentences make the abstract unreadable. Line 18: better wording would be "..presented by means of the physicochemical activation of coffee waste..." Line 20: chromium and (III) should be together Line 21: "the initial concentration of chromium..." is misplaced and should be moved, possibly as a separate sentence. Line 29: The series includes electroplate and electroplating redundancy. Does whiskies refer to singular whisky or alcohol in general? Should "in others" be "and others"? Line 31: "susceptible" should better refer to the water being contaminated, but it is used to describe heavy metals without the use of appropriate commas. Line 33: "Mainly from various industries" is extremely vague. Line39-44: Run-on sentences and several missing punctuation marks make the paragraph unreadable

Response 1: The wording of paragraphs 12-15 was changed to imply what coffee waste is and what are the drawbacks.The oxidation state of chromium Cr (III) was placed.Paragraph 23 describes why the initial concentration of chromium (III) is not significant, however, this idea was not omitted since it is of great importance because it is a variable of the experimental design.Paragraphs 33-39 best describe the main sources that generate wastewater with heavy metals, emphasizing chromium.

Point 2: The "mufla" in Line 74 should be refered to as a box furnace or kiln. The model of the box furnace should be included.

Response 2: On line 81 it is specified that a muffle furnace is used as well as the model. Point 3: The parameters from Table 1 are fine, but possibly have the pH range a bit wider. If not, provide justification of the small range of pH 1-3. And the caption of Table 2 is mismarked as Table 1 Response 3: In line 94 the reason for the pH range used is mentioned, in addition a Prevalence Existence Diagram (figure 1) was added to justify it.Table numbering corrected. 

Point 4: With pH being determined as the parameter with the strongest correlation to chromium removal from the experimental design, would there be a futher investigation with expanding the pH range?

Response 4: According to the figure 1 that was added, extending the pH range would cause a precipitation of chromium (III) hydroxide, because increasing the pH decreases its solubility

 Point 5: I don't quite understand the referral o the caption for Figure 3, 4, and 5. Possibly rewrite them as full sentences

Response 5: Now there are figures 5, 6 and 7. Line 145 mentions the equations of the models of the isotherms. The figures are the representations of the data adjusted to the mathematical models which is explained in line 150

Reviewer 2 Report

nmmnv

Reviewer comments to the Author:

Reviewer:

COMMENTS FOR AUTHORS

Manuscript ID: chemistry-678504

Title: Preparation of Activated Carbon from Coffee Waste as an Adsorbent for the Removal of Chromium (III) from Water. Optimization for an Experimental Box- Behnken

Article Type: Research Article

As reviewer of your paper I have to thank you for submitting this paper for Chemistry, but regret to advise you that it is not acceptable for publication in its present form, and will require minor revision before it could be considered for publication.

I would be happy to consider a rewritten and resubmitted paper. In revising your paper you should consider all the points. When resubmitting your paper you must provide a point by point list of your response to the reviewer. This manuscript is well written. Results in this manuscript are worthy but still some of the statements and minor revision should be made before this manuscript can be accepted. Followed are some specific comments:

In Introduction, through a few sentences, give a brief description of the earlier experiments and resoults of some other scientists: removal of chromium (III) by coffee waste and activated carbon from coffee waste used for adsorption of Chromium(III)

Author Response

Point 1: In Introduction, through a few sentences, give a brief description of the earlier experiments and resoults of some other scientists: removal of chromium (III) by coffee waste and activated carbon from coffee waste used for adsorption of Chromium (III)

Response 1: New bibliographical references were added in the introduction, which mention current published techniques performed with different adsorbent materials

Reviewer 3 Report

The article “Preparation of Activated Carbon from Coffee Waste as an Adsorbent for the Removal of Chromium (III) from Water. Optimization for an Experimental Box- Behnken Design “ is an interesting and fits the subject of Chemistry. The authors explained Coffee erase is an organic waste consisting of the ground, roasted and processed grain for the extraction of coffee, being of great volume the waste obtained, which, if not treated and preserved for a long time, emanates unpleasant aromas and becomes an optimal means for the proliferation of flies, and other pests. Activated carbon has the characteristic of being a material that has a large surface area, because of this it is used in adsorption, which refers to the retention of atoms, ions and molecules on its surface. In this paper the production of activated carbon is presented using the waste of coffee by means of physicochemical activation, by calcining at 900 °C and subsequent activation with sulphuric acid, as well as the Box-Behnken design of three factors for chromium adsorption optimization (lll). It is determined that the optimal conditions for performing the adsorption are pH=3 and a contact time of 140 min, the initial concentration of chromium is not significant for adsorption. Under the optimal adsorb conditions 96 %, thus an application is given to the erasure of coffee for the production of activated carbon which serves as the adsorbent agent applied to the removal of chromium (lll) aqueous. The area of activated carbon obtained is 13657.89±251.09 m2/g determined by the acetic acid adsorption isotherm method. The result analysis is accurate and adequate. The authors, using the appropriate equipment, have thoroughly investigated the issue.

Therefore, the manuscript can be recommended for publication in Chemistry, but after a minor corrections. The following points should be taken into account:

What was the structure of the pores of the adsorbent obtained from coffee with such a large specific surface? It would be good to confirm the specific surface obtained in this way by other methods of analysis, eg ASAP. A very good supplement to this publication would be to show the forms of Cr (III) ions in the studied pH range using the widely available MEDUS program. Novelty elements should be better highlighted in the introduction. Papers should be cited in Introduction section; for example:

Metal Ions Removal Using Nano Oxide Pyrolox™ Material Nanoscale Research Letters Volume 12, (2017) Article number 95

The effect of hydrothermal modification of titanium phosphate on the adsorption affinity towards cadmium ions Physicochemical Problems of Mineral Processing, 55 (6) 2019, 1568-1576

Author Response

Point 1: What was the structure of the pores of the adsorbent obtained from coffee with such a large specific surface? It would be good to confirm the specific surface obtained in this way by other methods of analysis, eg ASAP. A very good supplement to this publication would be to show the forms of Cr (III) ions in the studied pH range using the widely available MEDUS program. Novelty elements should be better highlighted in the introduction.

Response 1: New bibliographical references were added in the introduction, which mention novel techniques (Lines 48-55)  In lines 121-124 the species of Cr (III) adsorbed with bae is discussed in the species distribution diagram (Figure 3), likewise the pH range is justified in lines 94-96 based on the diagram of predominant existence for Cr (III), built in the MEDUSA software

Round 2

Reviewer 1 Report

After thorough revisions made by the authors, this manuscript is recommended for publication.